# 1-Bit Quantization Meets Structured Pruning: Towards Extreme Compression of Large Language Models

## Abstract

Large language models (LLMs) have achieved remarkable success across a wide range of AI applications. However, their massive parameter scales pose substantial challenges for deployment in practice. Quantization, a widely adopted compression technique, reduces parameter precision to as low as 1 bit, substantially shrinking the size and storage footprint of LLMs. While existing 1-bit quantization methods have reached the theoretical lower bound of bit-width, they remain confined to low-level quantization and fail to fully exploit structured redundancy for further compression. This is because previous works mainly focus on element-wise weight saliency and overlook the structured distribution of the weight saliency map. As a first attempt, this paper explores a unified framework that integrates structured pruning with 1-bit quantization, leveraging the strengths of both approaches for more effective compression. To this end, we introduce a novel Structured Saliency Score metric to identify which structured units should be pruned or quantized within the LLMs. We showcase that the proposed metric can effectively coordinate the synergy between quantization and pruning with a theoretical analysis. Extensive experiments on diverse LLMs and benchmarks demonstrate that our approach not only surpasses existing binarization-based methods but also achieves memory savings while maintaining competitive performance.

## 1 Introduction

Large language models (LLMs) have revolutionized artificial intelligence with remarkable capabilities in natural language processing and beyond, including applications in vision-language and multimodal reasoning (Zhu et al., 2024; Wang et al., 2024). However, the billions of parameters contained in modern LLMs lead to severe storage demands and computational overhead (Zhao et al., 2023), posing significant challenges for efficient deployment, especially in resource-constrained environments. As a result, developing effective compression techniques for LLMs has become a critical research direction.

Existing approaches to LLM compression largely fall into three categories: knowledge distillation (Gu et al., 2023; Agarwal et al., 2023; Ko et al., 2024; Padmanabhan et al., 2024), structured or unstructured pruning (An et al., 2024; Song et al., 2024; Ashkboos et al., 2024; Zhong et al., 2024; Ma et al., 2023; Sun et al., 2023; Frantar & Alistarh, 2023), and quantization (Frantar et al., 2022; Chee et al., 2024; Egiazarian et al., 2024; Zhang et al., 2024). Among them, quantization has emerged as one of the most effective strategies, substantially reducing model size and GPU memory footprint. In particular, binarization has recently gained momentum, compressing weights to just 1 bit per parameter.

Despite impressive results, recent binary post-training quantization (PTQ) methods (PB-LLM (Yuan et al., 2024), BiLLM (Huang et al., 2024a), STBLLM (Dong et al., 2025), ARB-LLM (Li et al., 2025)) remain confined to fine-grained weight discretization and largely ignore **structured saliency**. For example, PB-LLM examines both magnitude- and Hessian-based criteria and empirically favors the magnitude heuristic. EasyQuant (Tang et al., 2023) instead employs an $n\sigma$ rule to select salient weights. Both lines of work overlook the distributional characteristics of saliency map since

they adopt the element-wise saliency metric. Although SpQR (Dettmers et al., 2023), BiLLM, STBLLM, and ARB-LLM recognize that salient weights exhibit structured patterns, their computation of structured saliency typically proceeds by element-wise scoring followed by within-structure summation. Such additive measures can obscure intra-unit heterogeneity and interactions, thereby failing to faithfully capture the true structured saliency. Importantly, they fail to address **structured redundancy** that persists even after binarization, limiting the full potential of extreme compression.

As shown in Figure 1a and 1b, Magnitude-based and Hessian-based methods produce indistinct structured patterns regarding salient weights of LLM units. To this end, we introduce the Structured Saliency Score (SSS), a unified metric that jointly identifies the weight saliency and structured redundancy. The result is a clear saliency map that guides both quantization and pruning (See Figure 1c). In addition, our theoretical analysis supports that the SSS metric can effectively synergize 1-bit quantization and structured pruning. These findings suggest that structured pruning and binarization are not competing strategies but rather complementary techniques that can be unified for more effective compression.

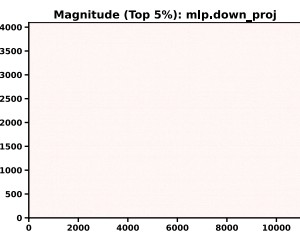 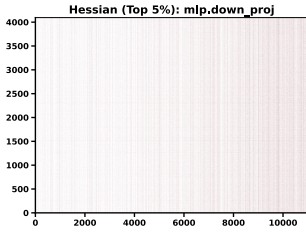 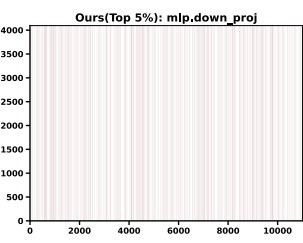

(a) Magnitude-based distribution    (b) Hessian-based distribution    (c) SSS-based distribution (Ours)

Figure 1: Distribution of top 5% salient weights in weight matrix. We take the `down_proj` layer of the MLP as an example and apply different criteria to evaluate weight saliency. In the figure, the horizontal axis corresponds to the column indices of the weight matrix and the vertical axis corresponds to the row indices. The top 5% of weights with the highest saliency scores are highlighted in red. (a) The magnitude criterion rarely reflects the structured regularity in the salient weights. (b) Using the Hessian criterion, the salient weights exhibit an indistinct structured pattern unfavorable to structured pruning. (c) Our SSS metric produces a clearer saliency map that enables structured pruning after quantization. (Zoom in if necessary)

In this work, we propose a unified framework that integrates structured pruning with 1-bit quantization. By leveraging both forms of redundancy—structured and precision—we push the boundaries of extreme LLM compression while preserving accuracy. Our main contributions are as follows:

- We propose the first framework that synergistically combines structured pruning with 1-bit quantization, exploiting their complementary strengths for extreme LLM compression.

- We design a Structured Saliency Score (SSS) metric that not only identifies salient weights but also accounts for their structured distribution, guiding pruning and quantization in a balanced and effective manner.

- Extensive experiments across diverse LLM architectures and benchmarks demonstrate that our method consistently outperforms state-of-the-art binarization approaches, achieving substantial memory savings while maintaining competitive performance.

## 2   RELATED WORK

**LLM Quantization.** Quantization reduces model parameters from full precision to lower-bit representations, thereby decreasing storage requirements and memory usage during inference. As an efficient model compression strategy, it has become one of the most widely used compression strategies for LLMs. Broadly, quantization approaches can be divided into Quantization-Aware Training (QAT) and Post-Training Quantization (PTQ).

QAT (Liu et al., 2024b; Chen et al., 2024; Wang et al., 2025) incorporates quantization into the training process, allowing the LLMs to adapt to low-bit representations during optimization. While

effective, QAT requires retraining large models, which is prohibitively costly for LLMs. PTQ methods (Frantar et al., 2022; Xiao et al., 2023; Lin et al., 2024; Huang et al., 2024b) instead quantize pre-trained LLMs directly, offering significantly lower computational overhead. For example, GPTQ (Frantar et al., 2022) introduces a one-shot PTQ method based on approximate second-order information, enabling 3–4 bit compression. AWQ (Lin et al., 2024) proposes protecting salient weights and searching for per-channel scaling factors based on activations. AMQ (Lee et al., 2025) proposes an automated framework that allocates different quantization bit widths across different layers, thereby achieving an optimal trade-off between accuracy and memory usage.

More recently, efforts have pushed toward 1-bit quantization for extreme compression. PB-LLM (Yuan et al., 2024), BiLLM (Huang et al., 2024a), STBLLM (Dong et al., 2025), and ARB-LLM (Li et al., 2025) allocate higher precision or refined search strategies to salient weights, achieving stronger performance under binarization. However, these methods remain restricted to fine-grained weight discretization. They do not address structured redundancy, which continues to exist even after binarization—a limitation our work aims to overcome.

**LLM Pruning.** Pruning eliminates redundant parameters in neural networks, and for LLMs, methods can be broadly categorized into unstructured and structured pruning.

Unstructured pruning removes individual weights from the matrix, achieving high sparsity while largely preserving accuracy. Examples include SparseGPT (Frantar & Alistarh, 2023), which applies the OBS technique (Hassibi et al., 1993) to prune GPT-family models, and Wanda (Sun et al., 2023), which removes weights with the smallest magnitudes scaled by their input activations. Alphapruning (Lu et al., 2024) leverages Heavy-Tailed Self-Regularization theory and spectral shape metrics to guide layer-wise sparsity allocation for unstructured pruning. However, unstructured pruning does not change matrix dimensions and thus yields limited inference acceleration. Acceleration benefits are often hardware-dependent, e.g., relying heavily on NVIDIA Ampere GPU support for $2:4$ or $4:8$ sparsity (Mishra et al., 2021).

Structured pruning, in contrast, removes entire components such as attention heads, neurons, or channels, which directly reduces both model size and computational cost, making it more deployment-friendly. SliceGPT (Ashkboos et al., 2024) employs embedding dimensionality reduction through weight slicing, while SlimLLM (Guo et al., 2025) introduces a structured framework that explicitly evaluates attention head and channel importance. Instruction-following pruning (Hou et al., 2025) employs a sparse mask predictor and enables input-dependent structured pruning by dynamically selecting structured units conditioned on user instructions.

Although structured pruning offers practical acceleration, existing methods operate on full-precision models. Their benefits have not yet been explored in conjunction with extreme low-bit quantization. Our work addresses this gap by unifying structured pruning and 1-bit quantization within a single framework, enabling both structured and precision-based compression for LLMs.

## 3 METHOD

### 3.1 1-BIT QUANTIZATION

1-bit quantization is a model compression method that extremely quantizes the parameters of LLMs from full-precision weight $\mathbf{W} \in \mathbf{R}^{n \times m}$ to binary values $\mathbf{Q} \in \{-1, 1\}$. To approximate the original weight, the 1-bit quantization objective function can be expressed as:

$$\arg \min_{\alpha, \mathbf{Q}} \|\mathbf{W} - \alpha \mathbf{Q}\|_F^2, \tag{1}$$

where $\alpha$ is a scaling factor. The optimal solutions for $\alpha$ and $\mathbf{Q}$ are given by $\alpha = \frac{\|\mathbf{W}\|_{l_1}}{n \times k}$ and $\mathbf{Q} = \mathrm{sgn}(\mathbf{W})$ (Rastegari et al., 2016), where $\mathrm{sgn}(\cdot)$ represents $\mathrm{sgn}(x) = \begin{cases} +1, & x \geq 0, \\ -1, & x < 0 \end{cases}$.

Since some salient weights are critical to the performance of the LLMs, many quantization approaches divide the original weights $\mathbf{W}$ into the salient and non-salient parts and adopt different quantization strategies for the two groups. Hence, the optimization problem can be reformulated as:

$$\arg \min \|\mathbf{W} - \alpha_{sal}\mathbf{Q}_{sal} \cup \alpha_{uns}\mathbf{Q}_{uns}\|_F^2. \tag{2}$$

In practice, it is critical to identify the salient parts within the weight matrix regarding 1-bit quantization. We analyze the saliency distribution within LLMs and discover that the weights' sensitive values are predominantly concentrated in specific columns. In other words, the granularity of saliency distribution is token-level. Therefore, we determine saliency through a per-column segmentation on the whole weight matrix, as shown in Figure 1.

## 3.2 STRUCTURED SALIENCY METRIC

**Point-Wise Weight Saliency.** Many quantization approaches, such as BiLLM, DB-LLM, STBLLM, and ARB-LLM, adopt the Hessian metric to measure the saliency score of individual weights. Another approach for point-wise saliency is Wanda (Sun et al., 2023):

$$\mathbf{S}_{i,j} = |\mathbf{W}_{i,j}| \|\mathbf{X}_j\|_2, \tag{3}$$

where $\mathbf{W}_{i,j}$ represents the value at $i$-th row and $j$-th column of the weight matrix, $|\cdot|$ is the absolute value operator, and $\|\mathbf{X}_j\|_2$ is the $l_2$-norm of $j$-th column input activation $\mathbf{X} \in \mathrm{R}^{r \times m}$.

In these methods, the saliency of a structured unit (e.g., columns of weight matrices) is typically computed as the sum of the saliency scores of its constituent elements. However, when a unit contains only a few highly influential elements and the majority contribute negligibly, summing element-wise saliency dilutes the impact of the critical elements. As a result, the overall saliency of the column is systematically overwhelmed, failing to reflect its true contribution. In Appendix A.2, we present a theoretical analysis that highlights the limitations of summation-based methods for evaluating structured saliency.

**Structured Saliency Score.** Beyond point-wise saliency, we aim to identify the structured distribution of the weight matrix, which seamlessly enables structured pruning after 1-bit quantization. To this end, we introduce the Structured Saliency Score (SSS) to evaluate structured saliency, which directly leverages structured relevance to determine structured saliency:

$$\mathbf{S}_j = \sigma(|\mathbf{W}_{:,j}|) \|\mathbf{X}_j\|_2, \tag{4}$$

where $\sigma(\cdot)$ denotes the calculation of the standard deviation. The rationale for our SSS metric is twofold: First, our SSS metric explicitly operates at a structural level, treating each weight column $\mathbf{W}_{:,j}$ as a cohesive unit. This approach allows us to assess the collective importance of an entire feature channel, moving beyond methods that simply aggregate the scores of individual weights within it. Second, our metric uses standard deviation, rather than magnitude, as a proxy for discriminative power. The underlying motivation is that a high standard deviation signifies large variations among the weights in a column. This implies that the corresponding feature channel applies a complex and specialized transformation, making it highly important. Conversely, a low standard deviation indicates that the weights are nearly uniform, suggesting the feature channel is less informative and therefore redundant.

Our proposed metric possesses two key properties: *1. Consistency with the number of salient elements. 2. Consistency with pruning loss.* Thus, the proposed metric can be applied not only to identifying salient parts in 1-bit quantization but also to assessing structured unit's salience in structured pruning. A formal mathematical proof of two key properties is presented in Appendix A.1. In addition, we conduct an ablation study in Table 6 to validate the design of Equation (4).

## 3.3 QUANTIZATION

**Non-Salient Part.** We use Equation (4) to divide the weight into salient $\mathbf{W}_{sal}$ and non-salient $\mathbf{W}_{uns}$ parts. For the non-salient part, a common approach is to further partition it according to the weight distribution and then quantize each set separately (Huang et al., 2024a; Fang et al., 2020; Zhou et al., 2017; Zhao et al., 2019). Thus, we define the quantization error for non-salient part as:

$$\arg \min_{\{p_\gamma\}_{\gamma \in \mathcal{G}}} \sum_{\gamma \in \mathcal{G}} \|\mathbf{W}_{uns}^{p_\gamma} - \alpha_{uns}^\gamma \mathbf{Q}_{uns}^\gamma\|_F^2, \tag{5}$$

where $\mathbf{W}_{uns}^{p_\gamma}$ denote the sub-set of non-salient part partitioned by the break-point $p_\gamma$. $\mathbf{Q}_{uns}^\gamma$ and $\alpha_{uns}^\gamma$ represent the binarized weights and the scaling factors, respectively. The detailed solution of break-points $\{p_\gamma\}_{\gamma \in \mathcal{G}}$ is provided in Appendix B.2.

**Salient Part.** For the salient parts, some methods finetune the quantization results by additionally introducing bias compensation (Liu et al., 2024a; Nagel et al., 2019; Li et al., 2021; Cai et al., 2020), which can be expressed as:

$$\mathbf{W}_{sal} = \hat{\alpha}_{sal}\hat{\mathbf{Q}}_{sal} + \mathbf{W}_{bias}. \tag{6}$$

At first stage, we optimize Equation (2) to obtain initial quantization results $\hat{\mathbf{Q}}_{sal}$ and $\hat{\alpha}_{sal}$. Then, $\mathbf{W}_{bias}$ is introduced as bias compensation while $\hat{\mathbf{Q}}_{sal}$ and $\hat{\alpha}_{sal}$ are fixed. Formally, this procedure is defined as follows:

$$\alpha_c, \mathbf{Q}_c = \arg\min_{\alpha_c, \mathbf{Q}_c} \left\| \mathbf{W}_{sal} - \hat{\alpha}_{sal}\hat{\mathbf{Q}}_{sal} - \alpha_c\mathbf{Q}_c \right\|_F^2, \tag{7}$$

where $\mathbf{W}_{bias} = \alpha_c\mathbf{Q}_c$. We use the approach in (Huang et al., 2024a) for optimization. Similar to non-salient part, we divide the salient weight matrix into sub-regions and quantize them separately.

**Total Bits Overhead.** Our compression scheme introduces overhead from two primary sources: the representation of weights with their associated parameters and mapping cost inherent to the grouping strategy. First, for the parameter bit, bias compensation (Equation (6)) results in an average storage cost per weight: $N_{bit} = 1 + r_{sal}$, where 1 is for the binary weight and $r_{sal}$ represents the proportion of salient weight. Incorporating with structured pruning, where $\varrho$ is the pruning ratio, the parameter cost becomes: $\tilde{N}_{bit} = (1 - \varrho) \cdot N_{bit}$. Second, our grouping strategy (Equation (5)) introduces a mapping cost. An indexing scheme (Chan & Ioannidis, 1998) is required to associate each weight with its corresponding group, which incurs a storage overhead of $N_{map} = \lceil \log_2(|\mathcal{G}|) \rceil$ bits, where $|\mathcal{G}|$ is the number of groups. Therefore, the overall bit cost of our method is:

$$N_{total} = \tilde{N}_{bit} + N_{map} = (1 - \varrho) \cdot (1 + r_{sal}) + \lceil \log_2(|\mathcal{G}|) \rceil. \tag{8}$$

Some studies (e.g., BiLLM and STBLLM) argue that the additional storage bits $N_{map}$ do not affect the acceleration of binary quantization and are therefore excluded from the total bit. In contrast, PB-LLM suggests that storage bits can be further compressed. In our work, all experiments explicitly report the additional storage bits to ensure a fair comparison.

### 3.4 STRUCTURED PRUNING

**Pruning Attention Layer**. Modern LLMs, such as the LLaMA family and Vicuna, employ Multi-Head Attention (MHA) (Vaswani et al., 2017), where the outputs of parallel attention heads are concatenated and linearly transformed by a final `output_projection` layer. We leverage this architecture for efficient structured pruning.

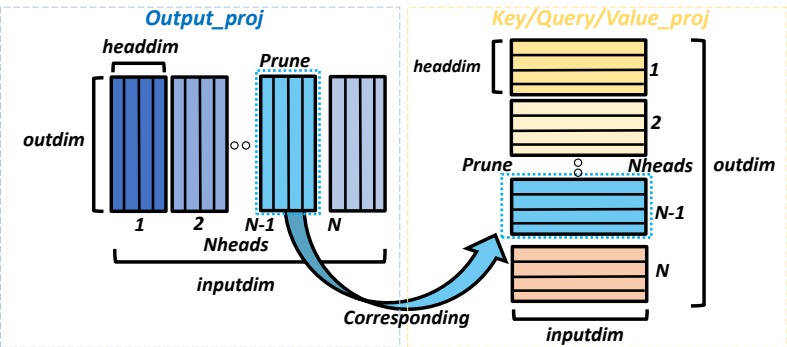

Figure 2: The detailed attention head pruning process. We denote the sub-block in the output projection layer corresponding to an attention head as a 'head'.

Instead of assessing each head's query, key, and value projections, we use the `output_projection` layer as a proxy for attention head importance. The rationale is that this layer directly reflects the contribution of each attention head to the final output. A sub-block of the `output_projection` matrix with a low saliency score indicates that its corresponding attention head has a negligible impact. Our method calculates the saliency score of each sub-block in the `output_projection` layer using Equation (22). Subsequently, we prune the low-saliency sub-block, which further allows for the removal of the corresponding attention head,

as shown in Figure 2. This principle extends naturally to models using Grouped-Query Attention (GQA) (Ainslie et al., 2023), such as LLaMA2-70B and LLaMA3-8B. For GQA, the pruning granularity shifts from individual heads to groups of heads.

**Pruning MLP Layer**. We apply a similar structured pruning principle to the MLP layers. In the common MLP architecture, the `gate_proj` and `up_proj` layers expand the network's intermediate dimension, which is then projected back by the `down_proj` layer. Analogous to our approach for attention heads, we use the `down_proj` layer as a proxy to determine the saliency of the intermediate neurons. We compute the saliency of `down_proj` layer by Equation (4) and prune the columns with the lowest scores. Subsequently, the corresponding rows from both the `up_proj` and `gate_proj` are removed.

### 3.5 Towards Extreme Compression: 1-Bit Quantization Meets Structured Pruning

This work achieves a synergy between 1-bit quantization and structured pruning by introducing the Structured Saliency Score (SSS) metric. Our framework leverages the complementary strengths of these two strategies for extreme LLM compression. The overall pipeline is illustrated in Figure 3.

**Structured Saliency Calculation.** First, we employ the proposed SSS criterion (Section 3.2) to evaluate the importance scores of all structural units within LLMs. As discussed in Section 3.4, in attention layers, we score attention heads by evaluating their corresponding sub-blocks in the `output_projection` matrix. In MLP layers, we evaluate intermediate neurons by assessing their corresponding columns in the `down_proj` matrix.

**Structure-Aware Quantization.** Based on the structured saliency map, we divide the weight matrices into salient and non-salient sub-regions, denoted by $\mathcal{G}$ and $\mathcal{V}$. We then binarize $\mathcal{G}$ and $\mathcal{V}$ seperately, where bias compensation is additionally applied to $\mathcal{G}$. In the subsequent update step, we apply block-wise error compensation (Frantar & Alistarh, 2023; Frantar et al., 2022) to preserve performance following post-training quantization.

**Post-Quantization Pruning.** Finally, given the pruning ratio, we prune structured units (i.e., attention heads and rows&columns of weight matrices) of the quantized model with low SSS score. We summarize the workflow of the compression framework and the pseudo code of implementation in Appendix B.1 and Appendix B.2, respectively.

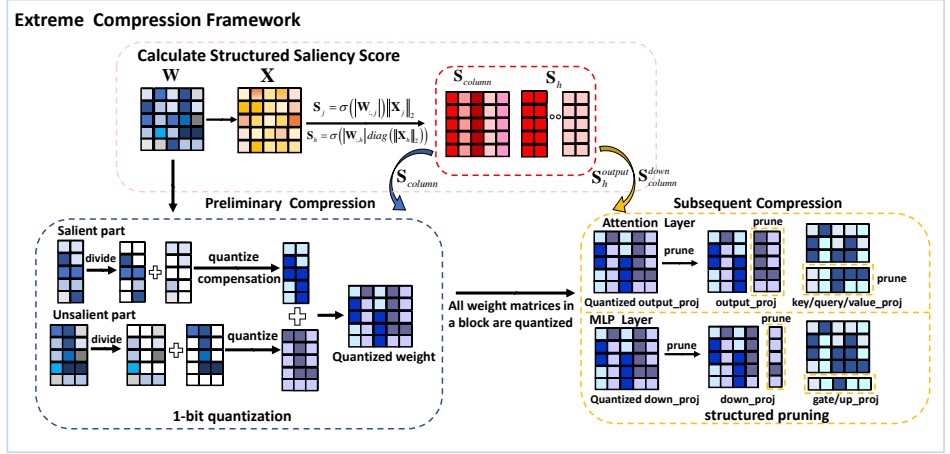

Figure 3: An overview of our extreme compression framework.

## 4 Experiments

### 4.1 Implementation Details

**Experimental Setup.** We evaluate the extreme compression method (Section 3.5) to various LLMs, including LLaMA series (Touvron et al., 2023a;b; Meta, 2024) and Vicuna (Chiang et al., 2023).

Experiments on the larger 65B and 70B models are conducted on 80GB NVIDIA A100 GPUs, while smaller models (e.g., 7B and 13B) are evaluated on 40GB NVIDIA L20 GPUs.

**Datasets and Evaluation.** The 128 samples from the C4 training dataset are randomly selected as the calibration set. We evaluate the perplexity of compressed models on the WikiText2 test dataset. Furthermore, we report the zero-shot performance on eight benchmark datasets: PIQA (Bisk et al., 2020); WinoGrande (Sakaguchi et al., 2021); HellaSwag (Zellers et al., 2019); ARC-e and ARC-c (Clark et al., 2018); OBQA (Mihaylov et al., 2018); BoolQ (Clark et al., 2019); RTE (Chakrabarty et al., 2021).

**Baselines.** We compare our proposed approach with other 1-bit quantization methods: PB-LLM (Shang et al., 2023), BiLLM (Huang et al., 2024a) , STBLLM (Dong et al., 2025), and ARB-LLM (Li et al., 2025). Other low-bit PTQ algorithms, such as GPTQ (Frantar et al., 2022), and PB-LLM (Shang et al., 2023) are also compared. Among these methods, some quantization methods use different storage bit widths. To ensure fair comparison, we conduct comparative experiments accordingly with the same storage bits. To compare quantization methods under the 1-bit storage bit, our method follows the grouping strategy of ARB-LLM, partitioning both the salient and non-salient parts into two groups. For the 2-bit storage setting, we make full use of the available storage bits by dividing the non-salient part into four groups. More details of experimental settings is provided in Appendix C.1-Appendix C.3.

Table 1: WikiText2 perplexity of quantized LLaMA family models.

| Method | Srorage Bits | 1-7B | 1-13B | 1-30B | 1-65B | 2-7B | 2-13B | 2-70B | 3-8B |
|---|---|---|---|---|---|---|---|---|---|
| FULL PRECISION | - | 5.68 | 5.09 | 4.10 | 3.53 | 5.47 | 4.88 | 3.31 | 6.13 |
| GPTQ | - | 129.19 | 20.44 | 13.01 | 8.66 | 52.22 | 23.63 | 8.78 | - |
| PB-LLM | 1.00 | 82.76 | 36.60 | 23.72 | 12.81 | 66.41 | 151.09 | 28.37 | 73.08 |
| BiLLM | 1.00 | 41.04 | 14.70 | 10.17 | 8.49 | 32.29 | 16.67 | 8.41 | 54.93 |
| ARB-LLM | 1.00 | 14.86 | 10.40 | 7.82 | 6.56 | 15.57 | 12.15 | 6.16 | 25.69 |
| OURS | 1.00 | **14.09** | **8.69** | **7.03** | **6.17** | **12.04** | **11.45** | **6.08** | **22.03** |
| STBLLM | 2.00 | 12.43 | 8.35 | 6.51 | 5.72 | 11.21 | 9.99 | 5.76 | 22.02 |
| OURS | 2.00 | **8.64** | **6.71** | **5.68** | **4.96** | **7.99** | **7.08** | **5.21** | **19.20** |

## 4.2 RESULTS ON LLMs

**Perplexity Results.** We evaluate the performance of various quantization methods on the LLaMA family (LLaMA1, LLaMA2, LLaMA3) under the block size of 128. The WikiText2 performance results are presented in Table 1. Since salient weights exist, some 1-bit quantization methods cannot strictly adhere to the target average of 1 bit per parameter. In contrast, our approach combines quantization with structured pruning, ensuring that the parameter bit-width truly reaches 1 bit

Table 2: Average bits per parameter of different quantization approaches.

| Method | LLaMA1 | LLaMA2 | LLaMA3 |
|---|---|---|---|
| GPTQ | 2.00 | 2.00 | 2.00 |
| PB-LLM | 1.70 | 1.70 | 1.70 |
| BiLLM | 1.09 | 1.08 | 1.06 |
| STBLLM | 1.09 | 1.08 | 1.06 |
| ARB-LLM | 1.09 | 1.08 | 1.06 |
| Ours | **1.00** | **1.00** | **1.00** |

as shown in Table 2. Moreover, our method achieves superior performance on WikiText2, outperforming not only non-1-bit approaches (e.g., GPTQ and PB-LLM) but also other 1-bit quantization methods. Moreover, for LLaMA3-8B, our method requires only 1-bit of additional storage while achieving performance comparable to STBLLM with 2 storage bits.

**Zero-shot Tasks Results.** To further evaluate the performance of different 1-bit quantization methods, we conduct zero-shot experiments on eight datasets as shown in Table 3. Here we present the results on LLaMA2-7B and LLaMA3-8B, while more results for other LLMs are available in Appendix C.4. For LLaMA2-7B, our method achieves the highest accuracy across all eight downstream tasks. Furthermore, the experimental results on the LLaMA3-8B model show that although all methods suffer from a certain degree of performance degradation, our method consistently surpasses the others. Notably, even at 1-bit storage overhead, our method still achieves better performance than STBLLM with 2 storage bits.

Table 3: Zero-shot performance of the quantized LLaMA family.

| Model | Method | Storage Bits | Winogrande | Piqa | Hellaswag | Arc-e | Arc-c | OBQA | BoolQ | RTE | Avg |
|-------|--------|------|------|------|------|------|------|------|------|------|------|
| | FULLPRECISION | - | 69.06 | 79.11 | 76.01 | 74.58 | 46.25 | 44.20 | 77.68 | 62.82 | 66.21 |
| | BiLLM | 1.00 | 53.04 | 58.32 | 36.50 | 37.21 | 22.78 | 28.60 | 57.25 | 50.54 | 43.03 |
| LLaMA2-7B | ARBLLM | 1.00 | 58.48 | 67.36 | 48.81 | 47.64 | 27.56 | 29.00 | 68.96 | 53.43 | 50.17 |
| | OURS | 1.00 | **61.64** | **70.51** | **55.18** | **53.54** | **29.61** | **31.00** | 67.52 | **57.40** | **53.30** |
| | STBLLM | 2.00 | 62.27 | 71.65 | 57.22 | 54.38 | 31.48 | 33.20 | 65.35 | **57.04** | 54.07 |
| | OURS | 2.00 | **63.93** | **75.30** | **66.91** | **63.89** | **36.26** | **39.00** | **69.14** | 53.43 | **58.48** |
| | FULLPRECISION | - | 73.24 | 80.74 | 79.16 | 77.57 | 53.24 | 44.80 | 80.98 | 68.59 | 69.79 |
| | BiLLM | 1.00 | 53.51 | 55.22 | 34.17 | 34.01 | 20.48 | 25.80 | 52.81 | 51.62 | 40.95 |
| LLaMA3-8B | ARBLLM | 1.00 | **58.96** | 64.04 | 46.84 | 47.01 | 28.24 | **31.20** | 67.03 | 53.07 | 49.55 |
| | OURS | 1.00 | 56.12 | **69.64** | **47.60** | **52.02** | **28.50** | **31.20** | 65.41 | 52.71 | **50.40** |
| | STBLLM | 2.00 | **58.72** | 58.11 | 54.24 | 37.16 | 24.66 | 32.20 | **66.91** | **56.32** | 48.54 |
| | OURS | 2.00 | 58.01 | **69.42** | **60.05** | **54.76** | **32.34** | **32.60** | 46.51 | 53.07 | **50.84** |

**Sub-1-bit Compression.** By integrating structured pruning with 1-bit quantization, our framework achieves extreme, sub-1-bit compression rates. We compare our approach with STBLLM, another state-of-the-art extreme compression method, under aggressive target bit rates of 0.8 and 0.7 bits per parameter. The results in Table 4 show that the performance of our method consistently surpasses STBLLM in this highly compressed regime.

Table 4: Zero-shot performance of sub-1-bit compression.

| Model | Method | Weight Bits | Winogrande | Piqa | Hellaswag | Arc-e | Arc-c | OBQA | BoolQ | RTE | Avg |
|-------|--------|------|------|------|------|------|------|------|------|------|------|
| | STBLLM | 0.8 | 67.80 | 73.12 | 64.73 | 57.66 | 32.42 | 39.00 | **67.68** | **58.48** | 57.61 |
| LLaMA1-13B | OURS | 0.8 | **68.67** | **75.35** | **67.50** | **59.34** | **35.24** | 37.20 | 67.65 | 55.23 | **58.27** |
| | STBLLM | 0.7 | 66.61 | 71.60 | 60.80 | **56.40** | 31.48 | **35.40** | 65.26 | **53.79** | 55.17 |
| | OURS | 0.7 | **66.63** | **73.39** | **61.84** | 54.46 | **33.79** | **35.40** | **68.01** | 53.07 | **55.78** |
| | STBLLM | 0.8 | 61.72 | 70.35 | 58.85 | 57.66 | 33.62 | 36.40 | **68.13** | **60.29** | 55.88 |
| LLaMA2-13B | OURS | 0.8 | **64.80** | **73.99** | **64.19** | **61.20** | **38.14** | **39.80** | 67.95 | 57.04 | **58.39** |
| | STBLLM | 0.7 | 57.38 | 66.32 | 47.12 | 46.09 | 26.54 | 31.00 | 62.32 | **53.07** | 48.73 |
| | OURS | 0.7 | **57.70** | **67.41** | **51.50** | **49.78** | **29.18** | **35.20** | **64.10** | 52.35 | **50.78** |

**Memory Comparison.** Memory usage is a key metric for evaluating compression methods. Thus, we compare the memory footprint of different approaches on LLaMA1-7B/13B and LLaMA2-7B/13B, as summarized in Table 5. We compare the overall memory footprint of compressed weight matrices for different methods. Our method leverages structured pruning to directly shrink the weight matrix, the group and column bitmap, thereby achieving a significant reduction in memory footprint. The detailed formulas for calculating the memory usage of each method are provided in the Appendix D.

Table 5: Memory (GB) comparison between FP16, BiLLM, STBLLM and Ours. FP16 denotes the overall memory required by the to-be-compressed weight matrices.

| Method | LLaMA1-7B | LLaMA1-13B | LLaMA2-7B | LLaMA2-13B |
|--------|------|------|------|------|
| FP16 | 12.06 | 23.63 | 12.06 | 23.63 |
| BiLLM | 2.24 | 4.41 | 2.22 | 4.39 |
| STBLLM | 2.43 | 4.77 | 2.41 | 4.75 |
| Ours | **2.15** | **4.23** | **2.15** | **4.24** |

### 4.3 ABLATION STUDIES

Table 6: Zero-shot performance of different saliency metric.

| Model | Method | Winogrande | Piqa | Hellaswag | Arc-e | Arc-c | OBQA | BoolQ | RTE | Avg |
|-------|--------|------|------|------|------|------|------|------|------|------|
| | WANDA | 56.91 | 67.57 | 44.62 | **53.28** | **30.20** | 32.40 | 64.65 | 51.99 | 50.20 |
| LLaMA1-7B | HESSIAN | 58.17 | 64.20 | 46.44 | 41.04 | 25.60 | 32.00 | 64.33 | 53.79 | 48.20 |
| | OURS | **59.35** | **68.72** | **54.68** | 51.26 | 28.67 | **36.60** | **67.68** | **58.84** | **53.23** |
| | WANDA | **67.88** | 75.98 | **66.16** | 61.15 | 35.32 | **38.80** | 69.30 | 54.15 | 58.48 |
| LLaMA1-13B | HESSIAN | 65.11 | 74.70 | 65.65 | 60.69 | 35.92 | 37.40 | 71.65 | **57.76** | 58.61 |
| | OURS | 67.48 | **76.12** | 65.76 | **63.93** | **36.69** | 37.40 | **72.05** | 57.04 | **59.55** |

**Different Saliency Metric.** Our saliency metric adopts a structured evaluation strategy, enabling direct assessment of the saliency of structured units. In contrast, prior 1-bit quantization approaches typically compute element-wise saliency scores such as Wanda and Hessian and then aggregate them to obtain the saliency of a structured unit, leading to the loss of structured information. To assess this difference, we conduct an ablation study to use different metrics and evaluated on LLaMA1-7B

and LLaMA1-13B. As shown in Table 6, our structured saliency criterion consistently outperforms additive metrics. These results further validate the effectiveness of our structured metric in capturing the saliency of structured units.

**Grouping Strategy.** Different quantization methods adopt distinct grouping strategies for salient and non-salient parts. BiLLM and STBLLM partition only the non-salient part into two and three groups, respectively, whereas ARB-LLM divides both salient and non-salient parts into two groups. In our ablation study, we validate our method to follow the same grouping schemes as these baselines. The results shown in Table 7 demonstrate that our approach consistently achieves superior performance when adopting the same grouping strategies as compared methods. Notably, even when the salient region is not partitioned, our method still outperforms ARB-LLM. This demonstrates that our approach does not rely on a grouping strategy but consistently delivers better performance.

Table 7: Different grouping strategies comparison.

| Method | Salient Part | Unsalient Part | LLaMA1-7B | LLaMA1-13B | LLaMA2-7B | LLaMA2-13B |
|--------|--------------|----------------|-----------|------------|-----------|------------|
| BiLLM | 1 | 2 | 41.04 | 15.20 | 32.29 | 16.67 |
| Ours | 1 | 2 | **14.94** | **9.70** | **15.18** | **11.84** |
| STBLLM | 1 | 3 | 12.43 | 8.35 | 11.21 | 9.99 |
| Ours | 1 | 3 | **9.61** | **7.21** | **9.26** | **7.79** |
| ARBLLM | 2 | 2 | 14.86 | 10.40 | 15.57 | 12.15 |
| Ours | 2 | 2 | **14.09** | **8.69** | **12.04** | **11.45** |

**Different Calibration Samples and Calibration Datasets.** Since both our criterion and the Hessian-based criterion rely on the input, we aim to further investigate whether our method can maintain robustness and consistently outperform other 1-bit quantization approaches under varying calibration sample number on LLaMA2-7B. In Figure 4a, as the calibration sample number increases, ARB-LLM exhibits a performance drop at 256 samples and BiLLM exhibits considerable performance fluctuations, whereas the performance of our method improves with increasing sample number, demonstrating strong robustness. Furthermore, we evaluate the influence of different calibration datasets on LLaMA2-7B, with experiments conducted on WikiText2 and C4 in Figure 4b and 4c. The results demonstrate that our method achieves stable performance across both WikiText2 and C4.

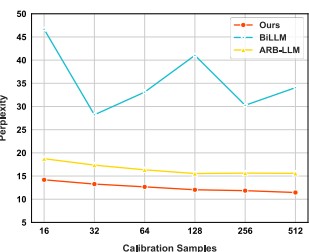 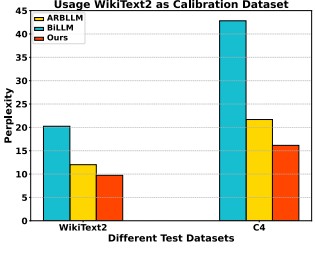 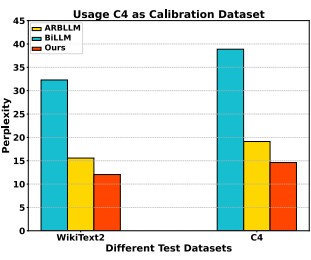

(a) Different Calibration Samples  (b) Calibration WikiText2  (c) Calibration C4

Figure 4: (a) The PPL results of different calibration samples. (b) Using WikiText2 training dataset as calibration dataset, the PPL results of WikiText2 and C4 test dataset. (c) Using C4 training dataset as calibration dataset, the PPL results of WikiText2 and C4 test dataset.

## 5 CONCLUSION

In this paper, we propose an extreme compression framework that integrates structured pruning with 1-bit quantization, enabling aggressive compression of LLMs and maximally eliminating redundancy. To this end, we propose a Structured Saliency Score (SSS) metric that serves as a unified criterion for both 1-bit quantization and structured pruning, enabling the identification of salient elements during quantization and the evaluation of structured unit saliency in structured pruning. Moreover, we evaluate our compression framework across various LLM architectures and benchmarks, comparing it against existing 1-bit quantization approaches. The experiment results demonstrate that our method consistently achieves superior performance to other 1-bit quantization approaches.

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

APPENDIX OVERVIEW

- **Section A:** Theoretical Analysis
- **Section B:** Method Implementation Details
- **Section C:** More Experiment Results
- **Section D:** Memory Computation Formulations
- **Section E:** LLMs Usage Statement

## A  THEORETICAL ANALYSIS

### A.1  PROOF OF PROPERTIES

**Property 1.** *Consistency with the number of salient elements.*

*Proof.* For an arbitrary threshold $\tau$, the number of salient elements in column $j$ can be obtained by:

$$C_j(\tau) = \sum_{i=1}^{n} \mathbf{1}\{\mathbf{S}_{i,j} \geq \tau\} = \sum_{i=1}^{n} \mathbf{1}\{|\mathbf{W}_{i,j}| \geq \tfrac{\tau}{\|\mathbf{X}_j\|_2}\}. \tag{9}$$

The weight distribution in LLMs can be well-approximated by independent Gaussian variable (Si et al., 2025; Thamm et al., 2022), i.e. $\mathbf{W}_{i,j} \sim (0, \sigma_g), i \in (1, \cdots, n)$. Therefore, the elements of the $j^{th}$ column $|\mathbf{W}_{i,j}|$ follow a approximated half-Gaussian distribution:

$$|\mathbf{W}_{i,j}| = \sigma_j Z = \sqrt{1 - \frac{2}{\pi}} \sigma_g Z, \tag{10}$$

where $Z$ denote a standard half-normal random variable (i.e., the non-negative part of the Gaussian distribution $\mathcal{N}(0,1)$), which is independent of the column index $j$. The scale parameter $\sigma_j = \sigma(|\mathbf{W}_{:,j}|)$ is naturally proportional to $\sigma(\mathbf{W}_{i,j})$ and $\sigma$ denotes the operation of computing the standard deviation. The probability of an element being salient can be expressed as:

$$\begin{aligned} p_j(\tau) &= \mathbb{P}\bigg(|\mathbf{W}_{i,j}| \geq \frac{\tau}{\|\mathbf{X}_j\|_2}\bigg) \\ &= \mathbb{P}\Big(Z \geq \tfrac{\tau}{\sigma_j \|\mathbf{X}_j\|_2}\Big) = \bar{F}\left(\tfrac{\tau}{\sigma_j \|\mathbf{X}_j\|_2}\right), \end{aligned} \tag{11}$$

where $\bar{F}(u) = \mathbb{P}(Z \geq u)$ denote the tail distribution function of the half-normal distribution. Since $\bar{F}(u)$ is a non-negative truncation of the symmetric Gaussian distribution, it retains the property of log-concavity. Consequently, the saliency probability $p_j(\tau)$ increases monotonically with the $\sigma_j \|\mathbf{X}_j\|_2$. The expected number of salient weights in column $j$ is given by:

$$\mathbb{E}[C_j(\tau)] = n \cdot p_j(\tau) = n\bar{F}\left(\tfrac{\tau}{\sigma_j \|\mathbf{X}_j\|_2}\right). \tag{12}$$

Equation (12) implies that $\mathbb{E}[C_j(\tau)]$ is also monotonically increasing with $\sigma_j \|\mathbf{X}_j\|_2$. Furthermore, we naturally define the column-level saliency metric as:

$$\mathbf{S}_j = \sigma_j \|\mathbf{X}_j\|_2 = \sigma(|\mathbf{W}_{:,j}|) \cdot \|\mathbf{X}_j\|_2. \tag{13}$$

Therefore, the number of salient element $\mathbb{E}[C_j(\tau)]$ monotonically increases with $\mathbf{S}_j$. Ranking columns by $\mathbf{S}_j$ is equivalent to ranking them by the expected number of salient elements, thereby we can conclude that our proposed metric is consistent with the number of salient elements.  □

**Property 2.** *Consistency with pruning loss.*

*Proof.* When the $j$-th column is pruned (i.e., approximated as zero), the output change can be expressed as:

$$\Delta \mathbf{Y} = \mathbf{Y} - (\mathbf{W} - \mathbf{W}_{:,j}\mathbf{e}_j^T)\mathbf{X} = \mathbf{W}_{:,j}\mathbf{X}_j^T, \tag{14}$$

and its Frobenius norm satisfies:

$$\|\Delta \mathbf{Y}\|_F^2 = \|\mathbf{W}_{:,j}\mathbf{X}_j^T\|_F^2 = \|\mathbf{W}_{:,j}\|_2^2 \cdot \|\mathbf{X}_j\|_2^2. \tag{15}$$

We compute the expectation of the output change after pruning:

$$\mathbb{E}(\|\Delta \mathbf{Y}\|_F^2) = \mathbb{E}(\|\mathbf{W}_{:,j}\|_2^2) \cdot \|\mathbf{X}_j\|_2^2. \tag{16}$$

$\mathbb{E}(\|\mathbf{W}_{:,j}\|_2^2)$ can be represented as:

$$\mathbb{E}(\|\mathbf{W}_{:,j}\|_2^2) = \mathbb{E}(\|\|\mathbf{W}_{:,j}\|\|_2^2) = \sum_{i=1}^n \mathbb{E}(|\mathbf{W}_{i,j}|^2) = n\sigma_j^2, \tag{17}$$

Therefore, according to Equation (16) and (17), we can obtain:

$$\mathbb{E}(\|\Delta \mathbf{Y}\|_F^2) = n\sigma(|\mathbf{W}_{:,j}|)^2 \cdot \|\mathbf{X}_j\|_2^2 \propto \mathbf{S}_j^2. \tag{18}$$

Therefore, this establishes $\mathbf{S}_j$ as a consistent and principled measure of structured saliency, making it well-suited as a criterion for structured pruning. □

## A.2 LIMITATION OF ELEMENT-WISE SUMMATION

A straightforward alternative to many salience metrics are directly summing the element-wise saliency scores within each column:

$$\mathbf{S}_j^{\text{sum}} = \|\mathbf{X}_j\|_2 \sum_{i=1}^n \|\mathbf{W}_{i,j}\|_2. \tag{19}$$

However, this element-wise summation metric does not maintain consistency with the number of salient elements. We can prove that element-wise summation is not monotonically aligned with the actual number of salient elements.

**Inconsistency with the number of salient elements.** By constructing a counterexample, we show that the element-wise summation criterion fails to align with the number of salient elements.

*Proof.* Considering two columns (a peaky column $\vartheta$ and a uniform column $\varepsilon$) with identical $l_2$ norm $\|\mathbf{X}_\vartheta\|_2 = \|\mathbf{X}_\varepsilon\|_2 = \delta$, the elements of the two columns respectively exhibit the following conditions:

$$column\ p: \quad \mathbf{W}_{0,\vartheta} = 1,\ \mathbf{W}_{1:n,\vartheta} = 0,$$
$$column\ u: \quad \mathbf{W}_{i,\varepsilon} = 1/\sqrt{n},\ \forall i.$$

Therefore, their corresponding summation metrics satisfy:

$$\mathbf{S}_\vartheta^{\text{sum}} = \delta < \mathbf{S}_\varepsilon^{\text{sum}} = \sqrt{n}\delta. \tag{20}$$

Equation (20) shows that column $\varepsilon$ contains more salient elements than column $\vartheta$. Nevertheless, $\exists\ t \in \left(\frac{\delta}{\sqrt{n}}, \delta\right]$, $t$ represents the threshold of salient elements. The number of salient elements can be calculate:

$$C_\vartheta(\tau) = 1 > C_\varepsilon(\tau) = 0, \tag{21}$$

where $C_j(\tau)$ is the number of salient elements in $j^{th}$ column. Equation (21) indicates that column $\varepsilon$ indeed contains more salient elements. Equation (20) and Equation (21) lead to conflicting conclusions. Thus, we can conclude that the element-wise summation fails to maintain consistency with the expected number of salient elements. □

## A.3 EXPAND PRUNING METRIC

In Equation (4), we only presented the criterion for determining the saliency of a single column. However, structured pruning in attention layer is typically performed at the attention-head level. Therefore, we extend this criterion to assess the saliency of attention heads and further prove that it remains consistent with pruning loss as follow:

$$\mathbf{S}_h = \sigma(|\mathbf{W}_{:,h}|diag(\|\mathbf{X}_{:,h}\|_F)), \tag{22}$$

Property 2. *Consistency with pruning loss.*

*Proof.* The pruning loss can be expressed as:

$$\Delta\mathbf{Y} = \mathbf{W}_{:,h}\mathbf{X}_{:,h}^T = \sum_{j \in h} \mathbf{W}_{:,j}\mathbf{X}_{:,j}^T. \tag{23}$$

and its Frobenius norm $\|\Delta\mathbf{Y}\|_F$ satisfies:

$$\|\Delta\mathbf{Y}\|_F = \|\sum_{j \in h} \mathbf{W}_{:,j}\mathbf{X}_{:,j}^T\|_F. \tag{24}$$

By the triangle inequality of the Frobenius norm, we can obtain the formulation:

$$\|\Delta\mathbf{Y}\|_F = \|\sum_{j \in h} \mathbf{W}_{:,j}\mathbf{X}_{:,j}^T\|_F \le \sum_{j \in h} |\mathbf{W}_{:,j}\mathbf{X}_{:,j}^T|_F = \sum_{j \in h} \|\mathbf{W}_{:,j}\|_F\|\mathbf{X}_{:,j}^T\|_F. \tag{25}$$

By the Cauchy–Schwarz inequality, we can obtain the Formulation (26):

$$\|\Delta\mathbf{Y}\|_F^2 \le \left(\sum_{j \in h} \|\mathbf{W}_{:,j}\|_F\|\mathbf{X}_{:,j}^T\|_F\right)^2 \le h\sum_{j \in h} \|\mathbf{W}_{:,j}\|_F^2\|\mathbf{X}_{:,j}^T\|_F^2 \tag{26}$$

Therefore,

$$\|\Delta\mathbf{Y}\|_F^2 \le h\sum_{j \in h} \|\mathbf{W}_{:,j}\|_F^2\|\mathbf{X}_{:,j}^T\|_F^2. \tag{27}$$

Formulation (27) establishes an upper bound on the pruning loss introduced by pruning attention heads. We define:

$$\tilde{\mathbf{S}}_h = |\mathbf{W}_{:,h}|diag(\|\mathbf{X}_{:,h}\|_F), \tag{28}$$

$$\|\tilde{\mathbf{S}}_h\|_F^2 = \sum_{j \in h} \|\mathbf{W}_{:,j}\|_F^2\|\mathbf{X}_{:,j}^T\|_F^2. \tag{29}$$

Then, we calculate the standard deviation of the Equation (28):

$$\sigma^2(\tilde{\mathbf{S}}) = \frac{1}{nh}\sum_{j \in h}(\tilde{\mathbf{S}}_{:,j} - \mu)^2 \asymp \frac{1}{nh}\sum_{j \in h}(\tilde{\mathbf{S}}_{i,j})^2 = \frac{\sum_{j \in h}\|\mathbf{W}_{:,j}\|_F^2\|\mathbf{X}_{:,j}^T\|_F^2}{nh}, \tag{30}$$

where $\mu$ is the average value of $|\mathbf{W}|$ and $\asymp$ denotes asymptotic equivalence. Therefore, combining the Formulation (26) and (30), we can obtain:

$$\|\Delta\mathbf{Y}\|_F^2 \le h\sum_{j \in h} \|\mathbf{W}_{:,h}\|_F^2\|\mathbf{X}_{:,h}\|_F^2 \asymp nh^2\sigma^2(|\mathbf{W}_{:,h}|diag(\|\mathbf{X}_{:,h}\|_F)). \tag{31}$$

Based on the Formulation (31), we can conclude that our metric in pruning attention heads determines the upper bound of the pruning loss. □

# B    METHOD IMPLEMENTATION DETAILS

## B.1    EXTREME COMPRESSION FRAMEWORK ALGORITHM DETAILS

The detailed algorithmic process of the extreme compression framework is provided in Algorithm 1. We traverse all weight matrices across in a layers and apply 1-bit quantization to them. During this process, we compute the saliency of each column as well as the saliency of the attention heads. Finally, for weight matrices in attention layers, pruning is conducted at the attention heads based on the saliency of heads, whereas for weight matrices in MLP layers, pruning is applied along both the rows&columns based on the saliency of columns.

---

**Algorithm 1** Extreme Compression Framework

---

**Input:** weight matrix $\mathbf{W}^l$ in a layer; calibration data $\mathbf{X}$; $\beta$ denotes block size;
$\mathcal{X}$ is the number of salient columns; $\lambda$ is hessian regularizer;
$\mathcal{L}$ is the set of weight matrices in a layer; $\varrho$ is pruning ratio.
**Output:** extreme compression weight matrices $\mathbf{W}_c$ .
$\mathbf{H} = 2\mathbf{X}\mathbf{X}^T$, $\mathbf{H}^c = \text{Cholesky}((\mathbf{H} + \lambda\mathbf{I})^{-1})$, $\mathcal{X} = 0$
Traverse the weight matrices in $\mathcal{L}$ :
**for** $l = 1$ **to** $|\mathcal{L}|$ **do**
   $\mathbf{S}_j^l = \sigma(|\mathbf{W}_{:,j}^l|)\|\mathbf{X}_j\|_2 \longrightarrow$ column saliency
   $\mathbf{S}_h^l = \sigma(|\mathbf{W}_{:,h}^l|\|\mathbf{X}_h\|_2) \longrightarrow$ attention head saliency
   **for** $g = 0, \beta$ **to** $(N-1)\beta$ **do**
      $column_s^l\{\cdot\} = \textbf{Salient}(\mathbf{W}_{,g:g+\beta}^l, \mathbf{S})$
      $\{p_v^*\}_{v\in\mathcal{V}} = \textbf{OptimalSplitSearch}(\mathbf{W}_{i,j\in column_s}^l)$
      $\tilde{\mathbf{W}}_1^l, \ldots, \tilde{\mathbf{W}}_{\mathcal{V}+1}^l = \textbf{Quant}(\mathbf{W}_{p_{v-1}^* \leq |w_{i,j}| \leq p_v^*, j \in column_s}^l) \longleftarrow p_0^* = 0, p_{\mathcal{V}+1}^* = +\infty$
      $\hat{\mathbf{W}}_1^l, \ldots, \hat{\mathbf{W}}_{\mathcal{V}+1}^l = \textbf{Compensation}(\tilde{\mathbf{W}}_1^l, \ldots, \tilde{\mathbf{W}}_{\mathcal{V}+1}^l)$
      $\hat{\mathbf{W}}_{sal}^l = \hat{\mathbf{W}}_1^l \cup \ldots \cup \hat{\mathbf{W}}_{\mathcal{V}+1}^l$
      $\{p_\gamma^*\}_{\gamma\in\mathcal{G}} = \textbf{OptimalSplitSearch}(\mathbf{W}_{i,j\notin column_s}^l)$
      $\hat{\mathbf{W}}_1^l, \ldots, \hat{\mathbf{W}}_{\mathcal{G}+1}^l = \textbf{Quant}(\mathbf{W}_{p_{\gamma-1}^* \leq |w_{i,j}| \leq p_\gamma^*, j \notin column_s}^l) \longleftarrow p_0^* = 0, p_{\mathcal{G}+1}^* = +\infty$
      $\hat{\mathbf{W}}_{unsal}^l = \hat{\mathbf{W}}_1^l \cup \ldots \cup \hat{\mathbf{W}}_{\mathcal{G}+1}^l$
      $\hat{\mathbf{W}}_{:,g:g+\beta}^l = \hat{\mathbf{W}}_{sal}^l \cup \hat{\mathbf{W}}_{unsal}^l$
      $\mathbf{E} = (\mathbf{W}_{:,g:g+\beta} - \hat{\mathbf{W}}_{:,g:g+\beta}^l)/\mathbf{H}_{g:g+\beta,g:g+\beta}^c$
      $\mathbf{W}_{:,g:g+\beta}^l = \mathbf{W}_{:,g:g+\beta}^l - \mathbf{E} \cdot \mathbf{H}_{g:g+\beta,g:g+\beta}^c$
      $\mathcal{X} = \mathcal{X} + \text{len}(column_s^l\{\cdot\})$
   **end for**
  **end for**
**if** $\mathbf{W}^{l\in\mathcal{L}}$ in an Attention layer **then**
   $\mathbf{W}_c^{l\in\mathcal{L}} = \textbf{AttPrune}(\mathbf{W}^{l\in\mathcal{L}}, \mathbf{S}_h^{output}, \mathcal{X}, \varrho, \mathcal{L})$
   **return** $\mathbf{W}_c^{l\in L}$
**else if** $\mathbf{W}^{l\in\mathcal{L}}$ in a MLP layer **then**
   $\mathbf{W}_c^{l\in\mathcal{L}} = \textbf{MLPPrune}(\mathbf{W}^{l\in\mathcal{L}}, \mathbf{S}^{down}, \mathcal{X}, \varrho, \mathcal{L})$
   **return** $\mathbf{W}_c$
**end if**

---

## B.2    DETAILED SUB FUNCTIONS PROCESS

Algorithm 2 provides the detailed implementation of all sub functions introduced in Algorithm 1, covering 1-bit quantization process (**Quant**), bias compensation (**Compensation**), partitioning salient and non-salient components with a two-group division as an example (**OptimalSplitSearch**), and the structured pruning procedure (**AttPrune and MLPPrune**).

---

**Algorithm 2** Pseudo Code of Implementation

---

**func Salient$(\mathbf{W}, \mathbf{X})$**
 1: $\mathbf{S}_j = \sigma(|\mathbf{W}_{:,j}|)\|\mathbf{X}_j\|_2$
 2: $column_s\{\cdot\} = topk_{largest}(\mathbf{S})$
 3: $e \leftarrow \infty$
 4: $n^* = 0$
 5: **for** $i = 0, 1, \ldots, len(column_s) - 1$ **do**
 6:    $\hat{\mathbf{W}}_1 = \mathbf{Quant}(\mathbf{W}_{:,j \in column_s[:i]})$
 7:    $\hat{\mathbf{W}}_2 = \mathbf{Quant}(\mathbf{W}_{:,j \notin column_s[:i]})$
 8:    **if** $\|\mathbf{W} - (\hat{\mathbf{W}}_1 \cup \hat{\mathbf{W}}_2)\|^2 < e$ **then**
 9:      $e = \mathbf{W} - (\hat{\mathbf{W}}_1 \cup \hat{\mathbf{W}}_2)\|^2$
 10:     $n^* = i$
 11:    **end if**
 12: **end for**
 13: **return** $column_s\{: n^*\}$

**func OptimalSplitSearch$(\mathbf{W})$**
 1: $e = inf$
 2: $p^* = 0$
 3: **for** $i = 0.1, 0.2, \ldots, 0.9$ **do**
 4:    $p = i \cdot \max(abs(\mathbf{W}))$
 5:    $\hat{\mathbf{W}}_1 := \mathbf{Quant}(\mathbf{W}_{|w_{i,j}| \leq p})$
 6:    $\hat{\mathbf{W}}_2 := \mathbf{Quant}(\mathbf{W}_{|w_{i,j}| > p})$
 7:    **if** $\|\mathbf{W} - (\hat{\mathbf{W}}_1 \cup \hat{\mathbf{W}}_2)\|^2 < e$ **then**
 8:      $e = \mathbf{W} - (\hat{\mathbf{W}}_1 \cup \hat{\mathbf{W}}_2)\|^2$
 9:      $p^* = p$
 10:    **end if**
 11: **end for**
 12: **return** $p^*$

**func Compensation$(\mathbf{W})$**
 1: $\tilde{\mathbf{W}} = \mathbf{Quant}(\mathbf{W})$
 2: $\mathbf{W}_{bias} = \mathbf{Quant}(\mathbf{W} - \tilde{\mathbf{W}})$
 3: $\hat{\mathbf{W}} = \tilde{\mathbf{W}} + \mathbf{W}_{bias}$
 4: **return** $\hat{\mathbf{W}}$

**func Quant$(\mathbf{W})$**
 1: $\alpha = \frac{\|\mathbf{W}\|_{l1}}{m}$
 2: $\hat{\mathbf{W}} = \alpha \cdot \text{sgn}(\mathbf{W})$
 3: **return** $\hat{\mathbf{W}}$

**func AttPrune$(\mathbf{W}, \mathbf{S}_h^{output}, \mathcal{X}, \varrho, \mathcal{L})$**

 1: $k = \frac{\mathcal{X}}{|\mathcal{L}| \times unit\_dim} + \varrho \times unit\_num$
 2: $id = topk_{smallest}(\mathbf{S}_h^{output}, k)$
 3: $\mathbf{W}_c^{l \in \mathcal{L}} = prune(\mathbf{W}^{l \in \mathcal{L}}, id)$
 4: **return** $\mathbf{W}_c^l$

**func MLPPrune$(\mathbf{W}, \mathbf{S}^{down}, \mathcal{X}, \varrho, \mathcal{L})$**

 1: $k = \frac{\mathcal{X}}{|\mathcal{L}| \times unit\_dim} + \varrho \times unit\_num$
 2: $id = topk_{smallest}(\mathbf{S}^{down}, k)$
 3: $\mathbf{W}_c^{l \in \mathcal{L}} = prune(\mathbf{W}^{l \in \mathcal{L}}, id)$
 4: **return** $\mathbf{W}_c^l$

---

## C  MORE EXPERIMENT RESULTS

### C.1  PRUNING RATIO ASSIGNING

Bias compensation for salient elements inevitably introduces additional bit overhead, thereby rendering the corresponding parameters as 2-bit representations. To achieve true "1 bit per parameter" quantization, we further apply structured pruning based on the number of salient columns in each layer. We provide the pruning ratios of structured units across all layers under the final 1-bit compression setting as Figure 5 shown.

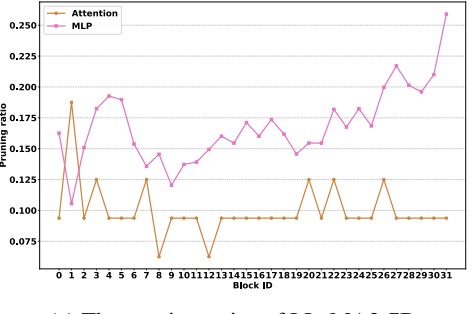

(a) The pruning ratios of LLaMA2-7B

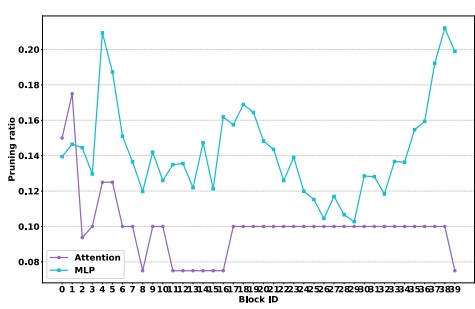

(b) The pruning ratios of LLaMA2-13B

Figure 5: The layer pruning ratios of LLaMA2-7B and LLaMA2-13B.

## C.2 Salient Column Number and Search Part Point Curve

**Salient Column Number Curve**. We illustrate the entire process of salient column search, with the results visualized in the form of error curves. Figure 6 illustrates the error curves of salient column search across a sub-block of `query_proj`, `up_proj`, `output_proj`, and `down_proj` layer. Clear optimal points can be observed in all cases, and the salient columns are predominantly concentrated in `output_proj` and `down_proj`.

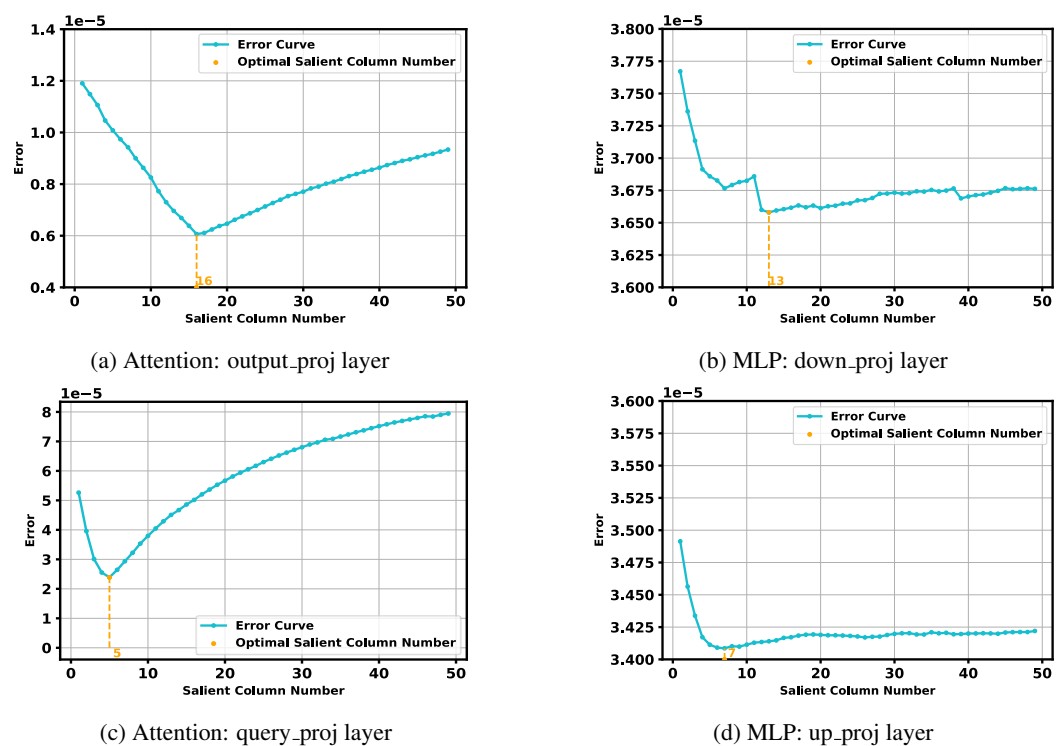

(a) Attention: output_proj layer        (b) MLP: down_proj layer

(c) Attention: query_proj layer        (d) MLP: up_proj layer

Figure 6: Salient column number curves of LLaMA2-7B.

**Searching Split Point Curve.** The process of identifying the optimal split point when both the salient and non-salient parts are further divided into two groups, as shown in Figure 7. The horizontal axis denotes the ratio between practical split point and the absolute of maximum weight value.

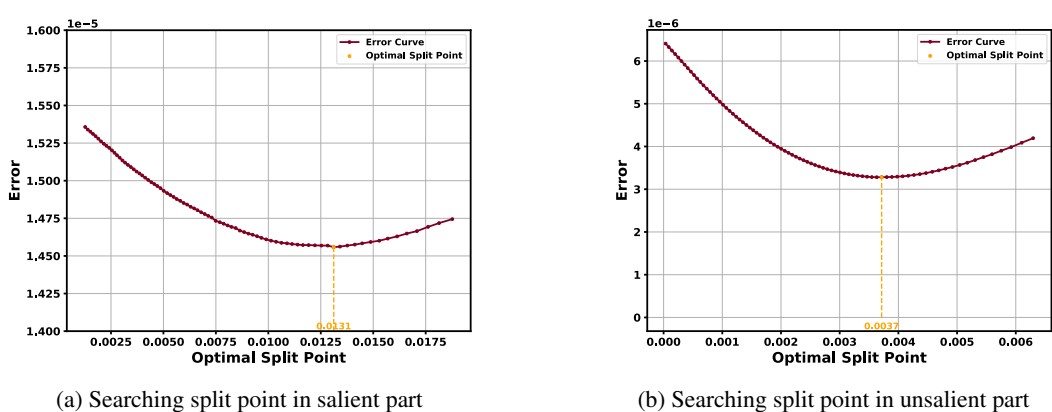

(a) Searching split point in salient part        (b) Searching split point in unsalient part

Figure 7: The curve of searching split point in a sub-block of output_proj layer.

## C.3 THE PARTITION RATIO BETWEEN SALIENT AND NON-SALIENT REGIONS.

Figure 8 illustrates the partition ratios of salient and non-salient parts in the `output_proj` and `down_proj` layers of LLaMA2-7B.

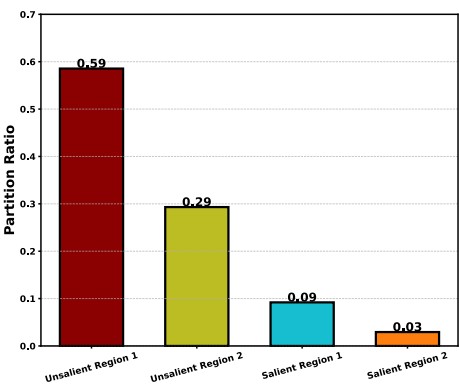

(a) The partition ratios of output_proj layer.

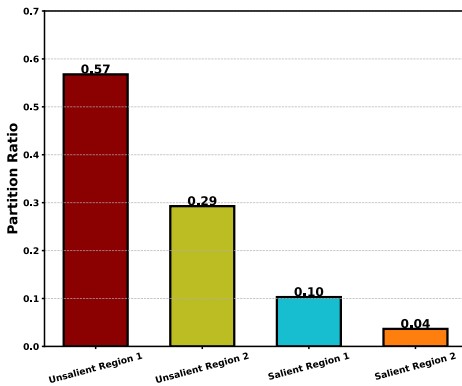

(b) The partition ratios of down_proj layer.

Figure 8: The proportion of each divided group in a sub-block of output_proj weight matrix.

## C.4 ZEROSHOT TASKS RESULTS ON VICUNA AND LLAMA FAMILY

**LLaMA Family**: The different 1-bit quantization methods are compared on more LLaMA families in Table 8.

Table 8: Zero-shot performance of different LLaMA families.

| Model | Method | Storage Bits | Winogrande | Piqa | Hellaswag | Arc-e | Arc-c | OBQA | BoolQ | RTE | Avg |
|---|---|---|---|---|---|---|---|---|---|---|---|
| | FULLPRECISION | - | 70.01 | 79.16 | 76.19 | 72.81 | 44.80 | 44.40 | 75.08 | 66.79 | 66.15 |
| | BiLLM | 1.00 | 52.72 | 59.30 | 35.87 | 37.88 | 25.17 | 27.40 | 61.65 | 52.35 | 44.04 |
| LLaMA1-7B | ARBLLM | 1.00 | **61.96** | 68.55 | 53.60 | **51.52** | **30.12** | 34.20 | **69.72** | **58.84** | **53.56** |
| | OURS | 1.00 | 59.35 | **68.72** | **54.68** | 51.26 | 28.67 | **36.60** | 67.68 | **58.84** | 53.23 |
| | STBLLM | 2.00 | 62.35 | 69.48 | 55.57 | 51.56 | 29.61 | 34.60 | 66.73 | 54.15 | 53.01 |
| | OURS | 2.00 | **64.01** | **74.81** | **66.32** | **62.25** | **38.05** | **36.80** | **70.49** | 51.99 | **58.09** |
| | FULLPRECISION | - | 72.77 | 80.14 | 79.09 | 74.79 | 47.70 | 44.80 | 77.95 | 70.40 | 68.45 |
| | BiLLM | 1.00 | 63.38 | 70.02 | 53.79 | 52.06 | 28.75 | 32.00 | 63.67 | 52.71 | 52.05 |
| LLaMA1-13B | ARBLLM | 1.00 | 67.09 | 73.72 | 62.09 | 60.98 | 34.13 | **38.00** | 70.70 | 55.60 | 57.79 |
| | OURS | 1.00 | **67.48** | **76.12** | **65.76** | **63.93** | **36.69** | 37.40 | **72.05** | **57.04** | **59.55** |
| | STBLLM | 2.00 | 67.32 | 75.14 | 67.89 | 59.05 | 35.07 | 39.40 | 71.10 | 58.84 | 59.23 |
| | OURS | 2.00 | **69.77** | **78.07** | **73.70** | **67.68** | **42.15** | **42.40** | **71.77** | **61.37** | **63.36** |
| | FULLPRECISION | - | 72.06 | 80.52 | 79.39 | 77.53 | 49.06 | 45.20 | 80.55 | 64.98 | 68.66 |
| | BiLLM | 1.00 | 57.93 | 63.11 | 41.37 | 45.08 | 26.02 | 30.60 | 64.71 | 52.71 | 47.59 |
| LLaMA2-13B | ARBLLM | 1.00 | **67.56** | **75.14** | 62.10 | 60.27 | 35.67 | **38.40** | 71.16 | **55.60** | 58.24 |
| | OURS | 1.00 | 67.48 | 74.65 | **64.37** | **65.66** | **37.37** | 36.20 | **73.98** | 54.15 | **59.23** |
| | STBLLM | 2.00 | 63.69 | 73.99 | 64.11 | 61.36 | 36.77 | 37.20 | **75.32** | 54.15 | 58.33 |
| | OURS | 2.00 | **70.09** | **76.66** | **72.69** | **70.08** | **42.83** | **41.40** | 69.85 | **56.68** | **62.53** |
| | FULLPRECISION | - | 75.69 | 82.26 | 82.62 | 78.96 | 52.99 | 48.00 | 82.66 | 67.15 | 71.29 |
| | BiLLM | 1.00 | 67.88 | 73.67 | 62.57 | 61.28 | 35.32 | 35.80 | 63.21 | 49.46 | 56.15 |
| LLaMA1-30B | OURS | 1.00 | 72.45 | **77.91** | **71.25** | **71.13** | **43.26** | 41.00 | 70.00 | **63.54** | **63.82** |
| | STBLLM | 2.00 | 71.11 | 78.02 | 72.88 | 69.65 | 42.24 | 41.60 | 79.85 | **65.34** | 65.09 |
| | OURS | 2.00 | **72.06** | **79.54** | **77.52** | **74.62** | **46.93** | **44.80** | **79.91** | 62.45 | **67.23** |
| | FULLPRECISION | - | 77.43 | 82.26 | 84.14 | 79.80 | 55.55 | 47.00 | 84.83 | 69.68 | 72.58 |
| | BiLLM | 1.00 | 70.96 | 76.22 | 66.92 | 65.19 | 36.60 | 40.40 | 79.08 | 53.43 | 61.10 |
| LLaMA1-65B | OURS | 1.00 | **74.98** | **79.65** | **75.79** | **73.36** | **47.18** | **41.20** | **80.58** | **59.93** | **66.58** |
| | STBLLM | 2.00 | 74.11 | 79.87 | 75.94 | 74.20 | 47.70 | 45.40 | **81.71** | 65.34 | 68.03 |
| | OURS | 2.00 | **74.74** | **80.96** | **79.00** | **76.47** | **50.26** | **45.20** | 81.35 | **70.76** | **69.84** |
| | FULLPRECISION | - | 77.98 | 82.81 | 83.78 | 81.02 | 57.34 | 48.80 | 83.73 | 67.87 | 72.92 |
| | BiLLM | 1.00 | 67.48 | 71.11 | 62.92 | 61.91 | 36.18 | 38.40 | 69.60 | 63.54 | 58.89 |
| LLaMA2-70B | OURS | 1.00 | **75.30** | **78.51** | **75.11** | **73.78** | **48.89** | **43.00** | **78.13** | **70.04** | **67.85** |
| | STBLLM | 2.00 | 74.98 | 79.16 | 76.94 | 75.72 | 52.47 | 43.60 | **80.95** | **66.79** | 68.83 |
| | OURS | 2.00 | **77.11** | **79.82** | **79.36** | **78.66** | **54.86** | **46.20** | 77.95 | 62.82 | **69.60** |

**Vicuna Family**: We further extend our proposed method to the instruction-tuned model Vicuna. As shown in Table 9, compared with other 1-bit quantization approaches, our method achieves the highest accuracy on eight zero-shot datasets using Vicuna-7B and Vicuna-13B. These results

demonstrate the strong generalization capability of our approach, which remains effective across different LLMs.

Table 9: The perplexity and zero-shot performance of the Vicuna family.

| Model | Method | Wikitext2 | Winogrande | Piqa | Hellaswag | Arc-e | Arc-c | OBQA | BoolQ | RTE | Avg |
|---|---|---|---|---|---|---|---|---|---|---|---|
| VICUNA-7B | FULLPRECISION | 6.78 | 69.53 | 78.02 | 73.76 | 71.34 | 45.82 | 45.00 | 80.95 | 63.90 | 66.04 |
| | BiLLM | 36.04 | 53.59 | 63.28 | 40.96 | 44.15 | 27.30 | 28.40 | 63.98 | 53.07 | 46.84 |
| | ARB-LLM | 18.71 | 57.46 | 68.06 | 51.90 | 51.01 | 31.14 | 29.00 | 72.29 | 52.71 | 51.70 |
| | OURS | 14.09 | 61.25 | 71.11 | 56.56 | 57.58 | 34.04 | 34.40 | 72.84 | 58.84 | 55.83 |
| VICUNA-13B | FULLPRECISION | 5.94 | 71.43 | 79.11 | 77.50 | 74.87 | 50.68 | 45.40 | 85.26 | 75.45 | 69.96 |
| | BiLLM | 38.73 | 55.80 | 64.42 | 44.95 | 44.74 | 28.58 | 29.80 | 66.61 | 56.68 | 48.95 |
| | ARB-LLM | 19.57 | 56.67 | 69.48 | 46.33 | 54.46 | 31.23 | 32.80 | 73.91 | 64.62 | 53.69 |
| | OURS | 10.09 | 67.09 | 74.86 | 64.72 | 66.58 | 39.16 | 39.20 | 81.07 | 66.06 | 62.34 |

## C.5 TIME COMPARISON

Our method involves not only quantization but also structured pruning, making it more sophisticated than other 1-bit quantization approaches. Nevertheless, compared with ARB-LLM, our compression process still requires less time.

Table 10: Time (s) comparison between ARB-LLM and Ours.

| Method | LLaMA1-7B | LLaMA1-13B | LLaMA2-7B | LLaMA2-13B |
|---|---|---|---|---|
| ARB-LLM | 3113 | 5146 | 3143 | 5117 |
| Ours | 3107 | 4763 | 2891 | 4771 |

## D MEMORY COMPUTATION

We provide the derivations of the memory computation formulas for our methods. For a weight matrix $\mathbf{W} \in \mathbf{R}^{n \times m}$, block size $\beta$ and the number of salient columns $s$, the memory $M^{\text{binary}}$ required can be formulated as:

$$M^{\text{binary}} = M^{\text{salient}} + M^{\text{salient}} + M^{\text{group-bitmap}} + M^{\text{salient-column-bitmap}}, \tag{32}$$

where $M^{\text{salient}}$ and $M^{\text{unsalient}}$ represent the memory computation of salient part and unsalient part respectively. Since all columns are divided into salient and non-salient parts, and the non-salient part is further partitioned into multiple groups, this inevitably introduces additional memory overhead $M^{\text{column-bitmap}}$ and $M^{\text{group-bitmap}}$. Moreover, $M^{\text{salient}}$ and $M^{\text{unsalient}}$ can be further formulated as:

$$M^{\text{salient}} = M^{\text{compensation}} + M^{\text{quantization-factor}}, \tag{33}$$

$$M^{\text{unsalient}} = M^{\text{1-bit}} + M^{\text{quantization-factor}}, \tag{34}$$

where $M^{\text{compensation}}$ denotes the additional memory overhead introduced by the compensation process, and $M^{\text{quantization-factor}}$ refers to the memory overhead incurred by the corresponding quantization factors, such as scaling factors and means. $M^{\text{1-bit}}$ is the memory overhead of 1-bit parameters. (Li et al., 2025) provide the memory computation formulas of BiLLM:

**- BiLLM**:

$$M^{\text{salient}} = 2nc + \lceil m/\beta \rceil \times 3n \times 16,$$

$$M^{\text{unsalient}} = n(m - s) + \lceil m/\beta \rceil \times 2n \times 16 \times 2,$$

$$M^{\text{group bitmap}} = nm,$$

$$M^{\text{column bitmap}} = m,$$

$$M^{\text{BiLLM}} = 2ns + \lceil m/\beta \rceil \times 3n \times 16 + n(m - s) + \lceil m/\beta \rceil \times 2n \times 16 \times 2 + nm + m. \tag{35}$$

Since STBLLM involves an unstructured pruning process, we assume a sparsity ratio of $\varrho$, then the proportion of nonzero elements is $\bar{\varrho} = 1 - \varrho$:

**- STBLLM**:

$$M^{\text{salient}} = 2nc + \lceil \bar{\varrho}m/\beta \rceil \times 3n \times 16,$$

$$M^{\text{unsalient}} = n(\bar{\varrho}m - s) + \lceil \bar{\varrho}m/\beta \rceil \times 2n \times 16 \times 3,$$

$$M^{\text{group bitmap}} = nm,$$

$$M^{\text{column bitmap}} = m,$$

$$M^{\text{STBLLM}} = 2ns + \lceil \bar{\varrho}m/\beta \rceil \times 3n \times 16 + n(\bar{\varrho}m - s) + \lceil m/\beta \rceil \times 2n \times 16 \times 2 + nm + m.$$

$$(36)$$

Since our method prunes either the $\varrho\%$ rows or the columns of the weight matrix (the proportion of the remaining structures is $\bar{\varrho} = 1 - \varrho$), it leads to two distinct cases of memory overhead $M^{\text{ours}}_{\text{column}}$ and $M^{\text{ours}}_{\text{row}}$:

**- Ours-column**:

$$M^{\text{salient}} = 2ns + \lceil \bar{\varrho}m/\beta \rceil \times 3n \times 16 \times 2,$$

$$M^{\text{unsalient}} = n(\bar{\varrho}m - s) + \lceil \bar{\varrho}m/\beta \rceil \times 2n \times 16 \times 2,$$

$$M^{\text{group bitmap}} = \bar{\varrho}nm,$$

$$M^{\text{column bitmap}} = \bar{\varrho}m,$$

$$M^{\text{ours}}_{\text{column}} = 2ns + \lceil \bar{\varrho}m/\beta \rceil \times 3n \times 16 \times 2 + n(\bar{\varrho}m - s) + \lceil \bar{\varrho}m/\beta \rceil \times 2n \times 16 \times 2 + n\bar{\varrho}m + \bar{\varrho}m.$$

$$(37)$$

**- Ours-row**:

$$M^{\text{salient}} = 2\bar{\varrho}ns + \lceil m/\beta \rceil \times 3\bar{\varrho}n \times 16 \times 2,$$

$$M^{\text{unsalient}} = \bar{\varrho}n(m - s) + \lceil m/\beta \rceil \times 2\bar{\varrho}n \times 16 \times 2,$$

$$M^{\text{group bitmap}} = n\bar{\varrho}m,$$

$$M^{\text{column bitmap}} = m,$$

$$M^{\text{ours}}_{\text{row}} = 2\bar{\varrho}ns + \lceil m/\beta \rceil \times 3\bar{\varrho}n \times 16 \times 2 + \bar{\varrho}n(m - s) + \lceil m/\beta \rceil \times 2\bar{\varrho}n \times 16 \times 2 + \bar{\varrho}nm + m.$$

$$(38)$$

# E  LLMs Usage Statement

We use large language model for grammar correction to enhance the clarity. We affirm that all conceptual and scientific contributions, including the formulation of the core ideas, the design of our proposed method, and the experimental framework, are entirely the original work of the authors and were developed without the use of generative AI tools.

