# OpenReview forum: "1-Bit Quantization Meets Structured Pruning: Towards Extreme Compression of Large Language Models"
_ICLR.cc/2026/Conference — ICLR 2026 Conference Withdrawn Submission_

### Official Review · Reviewer_oPsg · 2025-10-17

**Soundness:** 2
**Presentation:** 1
**Contribution:** 2
**Rating:** 2
**Confidence:** 4

**Summary:**

The paper proposes a framework combining 1-bit quantization and structured pruning for LLM compression. A Structured Saliency Score (SSS) is introduced to guide both pruning and binarization. Experiments on LLaMA and Vicuna show reduced memory usage and competitive perplexity/accuracy compared to prior 1-bit quantization methods.

**Strengths:**

1. The paper addresses extreme compression and presents the idea of combining quantization with structured pruning, which is a somewhat new framing.
2. The experimental setup is standard and covers several LLaMA and Vicuna models, and the paper also provides some mathematical proofs to support the proposed metric.
3. The paper is organized in a standard way, and the key ideas can be followed, though the writing could be clearer.
4. The general problem of efficient LLM compression is important and timely, which gives the paper some relevance despite limited technical depth.

**Weaknesses:**

1. The combination of 1-bit quantization and structured pruning seems mostly incremental. The paper claims synergy between 1-bit quantization and structured pruning, but fails to justify or demonstrate this. Beyond SSS, what design choices specifically enable this integration to outperform naive combinations?
2. Table 1 reports only “storage bits,” but not the total bit cost, and “storage” is also misspelled. If the method uses 1-bit quantization plus 2 bits of storage overhead, the effective cost is closer to 3 bits, yet the paper does not compare against any 3-bit baselines.
3. Table 2 claims 1.00 bit per parameter, but it is unclear how this is computed.
4. No evidence of real-world inference speedup is provided. Given the added complexity of bitmaps, it remains unclear whether the method would actually slow down inference in practice rather than accelerate it.
5. Memory savings are not compared against low-bit quantization methods (e.g., 2-bit or 3-bit). Current results suggest the compression ratio is closer to 2-bit baselines rather than “extreme” compression.
6. The experiments use older models (LLaMA-1/2/3 and Vicuna) rather than more recent LLMs, which limits impact.

**Questions:**

Please refer to the weakness section.

---

### Official Review · Reviewer_ubH1 · 2025-10-26

**Soundness:** 3
**Presentation:** 3
**Contribution:** 3
**Rating:** 4
**Confidence:** 5

**Summary:**

The authors of this manuscript propose a unified framework for extreme LLM compression that integrates 1-bit quantization with structured pruning. They introduce a novel metric, the Structured Saliency Score (SSS), which is designed to synergistically identify salient columns for quantization and unimportant structured units (e.g., attention heads, MLP neurons) for pruning. The stated goal is to push compression into the sub-1-bit regime by simultaneously exploiting both parameter precision and structural redundancy in a single, unified process.

**Strengths:**

1- The paper is well-written, and the core idea of a unified saliency metric for both quantization and pruning is easy to follow.

2- The accuracy and perplexity improvements reported, especially in the sub-1-bit regime, appear significant compared to other 1-bit quantization baselines like STBLLM and ARB-LLM.

3- The SSS metric is a novel contribution that is well-motivated and includes a theoretical analysis in the appendix to support its design.

**Weaknesses:**

1- The primary motivation for both structured pruning and 1-bit quantization is inference acceleration, yet the paper provides no end-to-end speedup benchmarks (e.g., in tokens/sec). This method combines two hardware-unfriendly techniques that typically require custom CUDA kernels for any practical speedup, as demonstrated by competing works like STBLLM [1]. Without these kernels and latency measurements on modern GPUs, the practical utility of the proposed method is unverified.

2- The evaluation of compression effectiveness could be strengthened. While the accuracy improvements are promising, it's unclear how these compressed models compare to smaller, dense models of an equivalent file size. To properly evaluate the practical trade-off, I would suggest plotting the model average accuracy vs. model size (in GB) for the LLaMA 3.x family, comparing the compressed models against their smaller, dense counterparts. I suggest LLaMA 3.x family of models since they span a wide range of LLM sizes (from 1B to 405B parameter count).

3- The current model selection focuses heavily on the LLaMA 1 and 2 families, which (based on my own experiments and the paper's results) appear to be less sensitive to compression. The paper's own data (Table 1, 3) shows a much more severe performance degradation for LLaMA3-8B. To better demonstrate the method's robustness, I suggest extending the evaluation to the broader LLaMA 3.x family (e.g., 1B, 3B, 13B, and 70B) and other recent models like the Gemma 3 family.

4- The comparison results are missing important benchmarks. The paper compares against other 1-bit quantization methods, but not against other combined or pruning-first approaches. How does this synergistic method compare to a simpler pipeline of applying a SOTA structured pruning method (like SliceGPT [2] or Wanda [3]) and then applying a 1-bit quantizer? This is a critical comparison needed to justify the complexity of the SSS framework.

---

[1] Dong et al., STBLLM: Breaking the 1-bit barrier with structured binary llms, ICLR 2025.

[2] Ashkboos et al., Slicegpt: Compress large language models by deleting rows and columns, ICLR 2024.

[3] Sun et al., A simple and effective pruning approach for large language models, 2023.

**Questions:**

1- Are there acceleration tools or custom CUDA kernels available for this 1-bit structured format? What are the measured end-to-end inference speedups (in tokens/sec) on recent GPUs (e.g., A100, H100) compared to the FP16 baseline and the STBLLM baseline?

2- How do the compressed LLMs from this method compare against smaller, dense models of the same final size (in GB)?

3- Can the authors extend the comparison to more recent and difficult-to-compress models, such as the LLaMA 3.x or the Gemma 3 family, to validate the method's scalability and robustness?

**Details Of Ethics Concerns:**

No concerns.

---

### Official Review · Reviewer_1uVj · 2025-10-27

**Soundness:** 2
**Presentation:** 1
**Contribution:** 3
**Rating:** 4
**Confidence:** 4

**Summary:**

This work presents a LLM compression method through a combination of structured pruning and 1 bit quantization. Experiments conducted on Llama-1/2/3 models show that they can be compressed to highly extreme (sub 1 bit) regimes while retaining significant performance.

**Strengths:**

* Combining Pruning and Quantization is an emerging area of research that could be very promising in the future
* Likewise, extreme (sub 3 bit), especially under 1 bit quantization is an underexplored region which could become more important as LLMs get larger and as test time scaling during inference gets more popular.
* 1 bit and sub one bit quantized models show a significant retention of the performance.

**Weaknesses:**

Overall the weakness of the paper can be lumped into three main sections.

**(Weakness 1: Writing)** A significant weakness of the paper is the poor writing, many aspects are hard to read and confusing. For example, beyond line 150, the variables are largely undefined, leaving the reviewer guessing as to their shapes and the set they are drawn from. This leads to particular confusion when trying to parse together how exactly the pruning and quantization aspects fit together, as well as the salience and non-salient portions of the weight. Furthermore, the method used to divide the weight into its salient and non salient portions is largely unspecified.

A good starting point would perhaps be to invert the order of the methods section by moving Section 3.5 to the front. For a non-comprehensive list of specific parts which are unclear please see the questions section.

**(Weakness 2: Models and Baselines)** Several key baselines were not considered. For example, PV-tuning[1] and OneBit[2] are very powerful one bit LLM compression methods. Perhaps the authors only consider the regime of one-shot model compression? But such a specific domain was not carved out in the paper. Additionally, for 2 bit, there are existing 2 bit one-shot methods such as QuIP#[3], NoWag-VQ[4], ICquant[5], that are much stronger baselines than STBLLM.

Likewise, the models that this method was evaluated on are rather old, with Llama 3 (the latest) being over a year old at the time of submission. Perhaps the authors could run additional experiments on newer models such as the Qwen 3 and Gemma 3 families? Furthermore the evaluation of model performance was done through wikitext2 perplexity and simple zero-shot tasks. These are largely no longer the primary metric to evaluate model performance, so it is difficult to evaluate the real world applicability of such a method. Perhaps more modern fewshot evaluations such as GPQA, MMLU, GSM8K, BBH could be run?

**(Weakness 3: Theoretical Results)**  The second theoretical claim is very misleading. What the authors refer to as the “pruning loss” is simply a commonly used layer wise proxy loss based on a hessian sketch obtained from the outer product of sample activations. Due to the size and complexity of an LLM, one cannot claim that such a proxy loss is wholly correlated with the performance loss of a compressed LLM (what ultimately matters). Indeed under certain regimes, this proxy loss will diverge from LLM performance, leading to different proxy losses being proposed[6][4], and different methods for obtaining a hessian sketch[7]. Additionally the entire theoretical results section (lines 199 through 203) is written in a non standard manner.



[1] Malinovskii, Vladimir, et al. "Pv-tuning: Beyond straight-through estimation for extreme llm compression." Advances in Neural Information Processing Systems 37 (2024): 5074-5121.

[2] Xu, Yuzhuang, et al. "Onebit: Towards extremely low-bit large language models." Advances in Neural Information Processing Systems 37 (2024): 66357-66382.

[3] Tseng, Albert, et al. "Quip#: Even better llm quantization with hadamard incoherence and lattice codebooks." arXiv preprint arXiv:2402.04396 (2024).


[4] Liu, Lawrence, et al. "NoWag: A Unified Framework for Shape Preserving Compression of Large Language Models." Second Conference on Language Modeling (2025).


[5] Xinlin Li, Osama Hanna, Christina Fragouli, and Suhas Diggavi. ICQuant: Index coding
enables low-bit LLM quantization. Second Conference on Language Modeling, 2025.

[6] Sun, Mingjie, et al. "A simple and effective pruning approach for large language models." arXiv preprint arXiv:2306.11695 (2023).

[7] Tseng, Albert, Zhaofeng Sun, and Christopher De Sa. "Model-Preserving Adaptive Rounding." arXiv preprint arXiv:2505.22988 (2025).

**Questions:**

* The work mentions a Hessian based saliency (such as in Fig 1b), how exactly is this calculated?
* How are the weights broken up into the Salient and nonsalient section (line 208 and 215\) is rather unclear. It seems that a break point is used, but how exactly is this identified?
* In the description of the quantization of the salient part (lines 216 to 229), the concept of a “bias” $\\mathbf{W}\_{\\mathrm{bias}}$ is introduced. The term bias is widely used to denote the additive bias in a linear layer/projection. Therefore, just to clarify, this “bias” is wholly unrelated to the existing nomenclature of a bias? If so, perhaps a better name could be found.
* The additional additive quantization $Q\_c$ for the salient weights, is this also one bit?
* The manuscript says that GPTQ style block-wise feedback is applied quantization (line 295). However the quantization error that this method minimizes seems to be just the simple Frobenius norm squared of the delta between the original and quantized weight (eqs 5 and 7). GPTQ style error compensation relies on a hessian sketch based proxy loss to introduce dependencies between the weights. With simpler proxy losses such as the Frobenius norm, error compensation makes no sense because the loss can be isolated into elementwise subproblems. So how exactly is this GPTQ feedback applied?
* The Wikitext 2 perplexities in Table 1 are computed at what context/sequence length? Since different context lengths impact the reported numbers.
* The existing body of work \[8\] claims that optimally ordering quantization and pruning should go pruning first and quantization second. This work does it the opposite way. Were any ablations done with prune first quantize second?
* In Table 1: the number of bits GPTQ uses is left as “-”. GPTQ was used to quantize how many bits? Also why is the entry for Llama 3 8B for GPTQ blank?
* For a standard model, such as Llama-2-7b, what fraction of the weights are pruned, what fraction is quantized, and what fraction is additionally quantized?
* How applicable is this method to MOE models?
* Was Llama-3 70B also evaluated? Table 1 only includes Llama-3 8B.
* The authors mention memory savings, was inference speedup also benchmarked?

\[8\] Harma, Simla Burcu, et al. "Effective interplay between sparsity and quantization: From theory to practice." *The Thirteenth International Conference on Learning Representations* (2025).

---

### Official Review · Reviewer_VgZh · 2025-10-30

**Soundness:** 3
**Presentation:** 3
**Contribution:** 2
**Rating:** 4
**Confidence:** 4

**Summary:**

This paper introduces a unified framework that integrates 1-bit quantization with structured pruning to achieve extreme compression of large language models. While prior binarization methods only focus on element-wise weight saliency, the authors propose a Structured Saliency Score (SSS) that captures both saliency and structured redundancy, enabling coordinated pruning and quantization. The framework identifies salient and non-salient structured units, applies binarization with bias compensation, and then prunes low-saliency structures such as attention heads and MLP neurons. Experiments on LLaMA and Vicuna models demonstrate that this approach consistently outperforms existing 1-bit quantization methods, achieving 1-bit per parameter storage, competitive perplexity, and strong zero-shot performance.

**Strengths:**

The paper offers a reasonable attempt to combine 1-bit quantization with structured pruning, which is a less explored direction. The proposed Structured Saliency Score provides a coherent way to connect the two techniques. The presentation is generally clear, and the experimental results suggest that the method can deliver competitive compression–accuracy trade-offs.

**Weaknesses:**

The experimental validation, while extensive, is limited to relatively dated model families such as LLaMA-1/2/3 and Vicuna. The absence of evaluations on more recent and stronger LLMs somewhat weakens the claim of broad significance. In addition, although the Structured Saliency Score is intuitively motivated, its theoretical novelty appears incremental relative to existing saliency-based heuristics. Finally, the practical deployment benefits are discussed only at a high level, with limited evidence of end-to-end acceleration or on-device efficiency.

**Questions:**

1. Could the authors provide results on more recent and stronger models (e.g., Qwen2.5, Qwen3) to strengthen the significance of the experimental validation?
2. The Structured Saliency Score is presented as a key contribution, but it seems closely related to existing saliency heuristics. Can the authors further clarify their theoretical novelty and explain how it fundamentally differs from prior metrics?
3. While memory savings are clear, the paper provides limited evidence of end-to-end acceleration or deployment gains. Can the authors include runtime benchmarks or device-level evaluations to demonstrate practical benefits?

---

### Official Review · Reviewer_aZvF · 2025-10-31

**Soundness:** 3
**Presentation:** 3
**Contribution:** 2
**Rating:** 2
**Confidence:** 4

**Summary:**

This paper tackles extreme LLM compression below ~1–2 bits per parameter without heavy retraining. It introduces a post-training pipeline that first binarizes weights to 1-bit, then prunes macro-structures using a Structured Saliency Score that blends per-column weight dispersion with activation energy. This paper reports lower perplexity and stronger zero-shot accuracy than prior 1-bit baselines.

**Strengths:**

1. This paper is the first framework that combines structured pruning with 1-bit quantization, exploiting their complementary strengths for extreme LLM compression.

**Weaknesses:**

1. This paper only conduct perplexity-based evaluations, including the zero-shot evaluations because the predictions in these benchmarks  are selected through the rank of the perplexity of each choice instead of through generation (sampling). These benchmarks are not enough to evaluate generative language models. I would suggest to include benchmarks like 5-shot MMLU and 8-shot GSM8K.
2. The paper motivates 1-bit quantization as a remedy for large memory consumption of LLMs, but it does not compare against smaller dense models with the same memory consumption. I would suggest to include comparison to small models. For example, Qwen3-0.6B-Base with fp16 (approx. 1.2 GB) and Qwen3-8B-Base with 1 bit quantization (approx. 16GB/12=1.33 GB, factor 12 is inferred from Table 5).

**Questions:**

NA

---

### Note · Authors · 2025-11-16

I have read and agree with the venue's withdrawal policy on behalf of myself and my co-authors.